# MME-Finance: A Multimodal Finance Benchmark for Expert-level Understanding and Reasoning

## Abstract

The remarkable capability of existing Multimodal Large Language Models (MLLMs) to understand general natural images have been extensively demonstrated in plentiful benchmarks. Nevertheless, the potential of MLLMs in finance domain remains to be fully explored. Financial images exhibit a wide range of variations, encompass intricate details, and demand professional expertise for proper interpretation, thereby posing a significant challenge for MLLMs in terms of their fine-grained perception and complex reasoning capabilities. To bridge this gap, we introduce MME-FINANCE, a novel benchmark designed specifically to assess MLLMs' performance in open-ended financial Visual Question Answering (VQA). Our benchmark consists of over 1,000 VQA pairs spanning a wide range of complex financial scenarios. We devise multi-tiered financial tasks tailored to the specific characteristics of the financial domain, aiming to comprehensively evaluate the perception, reasoning, and cognition capabilities of MLLMs. Furthermore, we employ a multimodal evaluation approach that incorporates visual data to score the model predictions, thereby aligning more closely with human judgment. Extensive experimental evaluations of 18 mainstream MLLMs reveal their limitations in financial tasks and provide insights to inspire further research.

## 1 Introduction

Multimodal Large Language Models (MLLMs), which equip the Large Language Models (LLMs) (Radford et al., 2019; Ouyang et al., 2022; Dai et al., 2022; Touvron et al., 2023) with the capability of visual understanding, have experienced a revolutionary advancement recently. Works including Flamingo (Alayrac et al., 2022), LLaVA (Liu et al., 2024c), CogVLM (Wang et al., 2023), Gemini (Team et al., 2023), and GPT-4o (Open-AI, 2024) have demonstrated intriguing capability to solve complex multimodal recognition and reasoning tasks. A reasonable and objective benchmark is of enormous significance in the success of MLLMs, which not only helps a better comparison of the performances of MLLMs but also provides valuable guidance for model optimization and real-world applications.

Early works of multimodal benchmarks, such as COCO Caption (Chen et al., 2015), GQA (Hudson & Manning, 2019), and Flickr30k (Young et al., 2014), have served as foundational resources for evaluating MLLMs. However, these benchmarks are task-specific, limiting the scope for fine-grained analysis of MLLMs' capabilities. More recent efforts, including MME (Fu et al., 2023), MMBench (Liu et al., 2023), and MM-Vet (Yu et al., 2023), have shifted focus towards general multimodal tasks. These benchmarks comprehensively evaluate the diverse capabilities of MLLMs, such as perception and reasoning, through a broader range of tasks. Alongside these general-purpose benchmarks, domain-specific benchmarks are rapidly emerging. For instance, in the medical field, benchmarks like GMAI-MMBench (Chen et al., 2024a) and Asclepius (Wang et al., 2024) have been developed, while in the autonomous driving domain, NuScenes-QA (Qian et al., 2024) and DriveLM-DATA (Sima et al., 2023) are advancing research. These benchmarks have significantly accelerated the progress of MLLMs within their respective industries.

In the financial field, there are many types of charts which contain a wealth of information that only professional financial experts can interpret. For MLLMs, they not only need to understand the basic visual information of financial images, but also need to combine financial knowledge to mine

important information. Therefore, it is challenging to comprehensively and professionally evaluate the financial capability of MLLMs. Benchmarks such as FINANCEBENCH (Islam et al., 2023) and CFBenchmark (Lei et al., 2023), are focusing on the evaluation of LLMs. To the best of our knowledge, there is no multimodal benchmark in the financial area. Hence, a professional financial multimodal benchmark is urgent for promoting the development of MLLMs.

To bridge this gap, we propose MME-FINANCE, a financial multimodal benchmark for MLLMs. We conducted extensive research on real-world financial application scenarios and selected 6 common types of financial charts, including candlestick charts, technical indicator charts, statistical charts, tables, documents, and mixed charts. Based on these images, we designed a hierarchical series of open-ended Question Answering (QA) tasks, ranging from general visual perception like Optical Character Recognition (OCR) tasks to complex cognitive tasks such as providing investment advice. These tasks evaluate the MLLMs' ability to visually comprehend and analyze expert financial domain knowledge. To ensure the quality of MME-FINANCE, we carefully designed the annotation pipeline and invited senior professional financial experts to conduct detailed verification of the answers. LLMs are employed for automated evaluation in MME-FINANCE. Considering the challenges of evaluating financial open-ended questions, we meticulously designed the evaluation process and selected performant models for assessment. Extensive experiments validate the effectiveness of our evaluation method. The experimental results of 18 MLLMs reveal that, while they demonstrate basic capability in analyzing financial images, existing MLLMs remain inadequate in meeting the requirements of fine-grained perception and complex cognition tasks. We summarize our major contributions as follows:

- We propose MME-FINANCE, a novel multimodal benchmark specifically designed to evaluate the capabilities of MLLMs in the financial domain. MME-FINANCE involves diverse financial image types and focus on various visual capabilities, such as perception, reasoning, and cognition, providing a comprehensive evaluation of MLLMs' performance in the financial domain.

- We introduce an evaluation approach of open-ended questions in the financial domain. By designing appropriate prompts for corresponding tasks and exploring evaluation methods combined with image information, we propose a novel evaluation strategy that has a high consistency with humans. The strategy can serve as a reference for evaluating MLLMs for other works.

- We conduct extensive evaluation on 18 MLLMs based on MME-FINANCE, revealing critical insights about the strengths and shortcomings of the current MLLMs in financial applications. The insights gained from this study provide a foundation for future research, guiding the development of more robust MLLMs capable of meeting the demands of complex financial tasks.

## 2 RELATED WORK

### 2.1 MLLMs

Recent advancements in LLMs (Radford et al., 2019; Brown, 2020; Ouyang et al., 2022; Touvron et al., 2023; Chiang et al., 2023) have catalyzed significant breakthroughs in MLLMs. Utilizing pre-trained LLMs allows researchers to circumvent the resource-intensive process of training models from scratch, thereby markedly reducing computational costs. By harnessing the cognitive capabilities of LLMs, MLLMs are adept at addressing diverse multimodal challenges. To facilitate alignment between different modalities, researchers have proposed several effective connectors. Models such as BLIP-2 (Li et al., 2023c), Mini-GPT4 (Zhu et al., 2023), Video-LLaMA (Zhang et al., 2023), and X-LLM (Chen et al., 2023) employ Q-Former for the alignment of visual and textual features, while the LLaVA series (Liu et al., 2024c;a) exploit MultiLayer Perceptrons (MLPs) for this purpose. Additionally, Flamingo (Alayrac et al., 2022) and CogVLM (Wang et al., 2023) incorporate supplementary modules to enhance interaction and fusion between visual and textual elements. Closed-source MLLMs, such as Gemini (Team et al., 2023), GPT-4V (Open-AI, 2023), GPT-4o (Open-AI, 2024), and Claude 3.5 Sonnet (Claude, 2024), demonstrate exceptional capabilities in visual understanding.

While these MLLMs demonstrate excellent performance in standard multimodal tasks such as image captioning (Vinyals et al., 2015) and Visual Question Answering (VQA) (Antol et al., 2015), their performance in specialized domains, particularly finance, remains relatively unexplored. Financial images generally present diverse content and necessitates specialized knowledge for interpretation, posing a substantial challenge for MLLMs.

## 2.2 Multimodal Benchmarks

MLLMs have demonstrated exceptional capabilities across various complex tasks. Objective and accurate quantification of these capabilities is essential for informing future development trajectories, making the establishment of comprehensive benchmarks significant for advancing MLLMs. Traditional multimodal benchmarks typically focus on single tasks, for instance, COCO Caption (Chen et al., 2015) and Flickr30k (Young et al., 2014) address captioning, while GQA (Hudson & Manning, 2019), VQAv2 (Goyal et al., 2017), and VizWiz (Gurari et al., 2018) pertain to VQA. Other benchmarks assess specific capabilities, such as TextCaps (Sidorov et al., 2020) and Tap (Yang et al., 2021) for scene text understanding, and VCR (Zellers et al., 2019) for commonsense reasoning. Subsequent benchmarks have expanded in both data volume and task categories. The MME benchmark (Fu et al., 2023) proposes a comprehensive assessment across 14 perception and cognition tasks, while MMBench (Liu et al., 2023) constructs over 3,000 multiple-choice image question pairs encompassing 20 abilities. SEED-Bench (Li et al., 2023a) and SEED-Bench-2 (Li et al., 2023b) further scale the sample sizes to 19,000 and 24,371 QA pairs from diverse scenarios, respectively. Collectively, these benchmarks provide thorough evaluations of MLLMs' capacities to tackle general multimodal challenges.

However, the performance evaluation of MLLMs in specific domains remains underexplored, particularly in finance. Existing benchmarks like FINANCEBENCH (Islam et al., 2023) and CFBenchmark primarily (Lei et al., 2023) assess LLMs rather than MLLMs.

## 3 MME-FINANCE

In this section, we introduce MME-FINANCE by first elaborating on the design philosophy of MME-FINANCE in 3.1, followed by a detailed description of the data collection in 3.2, data annotation in 3.3, and the statistics of MME-FINANCE in 3.4. Finally, we expound on the evaluation method of MME-FINANCE in 3.5.

## 3.1 Hierarchical Ability Levels of MME-FINANCE

The abilities of MLLMs can be divided into three categories: visual understanding, logical reasoning, and complex cognition. MME-FINANCE references these categories and organizes a three-tier ability structure. Specifically, in MME-FINANCE, we define perceptual ability as the low-level capacity for extracting and interpreting visual information from images. This foundational ability supports other advanced capabilities. To evaluate the perceptual ability, MME-FINANCE employs tasks such as image captioning, OCR, entity recognition, and spatial awareness. As a middle-level ability, reasoning encompasses financial-related numerical reasoning. MME-FINANCE evaluates this ability through tasks involving both estimated and accurate numerical calculations. Cognition is considered as a high-level ability, which requires integrating perceptual and reasoning skills with domain-specific financial knowledge to generate reasonable answers. The corresponding tasks, typically complex and requiring expert-level financial insight, include reason explanation, risk warning, investment advice, and financial knowledge QA. It should be noted that some cognitive tasks are insufficient to answer based solely on image information. For such tasks, MME-FINANCE provides additional background information retrieved via web searches, supplementing the images and questions. These tasks require MLLMs to synthesize both image content and background information to derive the correct answers. Additionally, to assess the capability to handle hallucinations in MLLMs, MME-FINANCE includes the not applicable task, which means the answer is not applicable for the question.

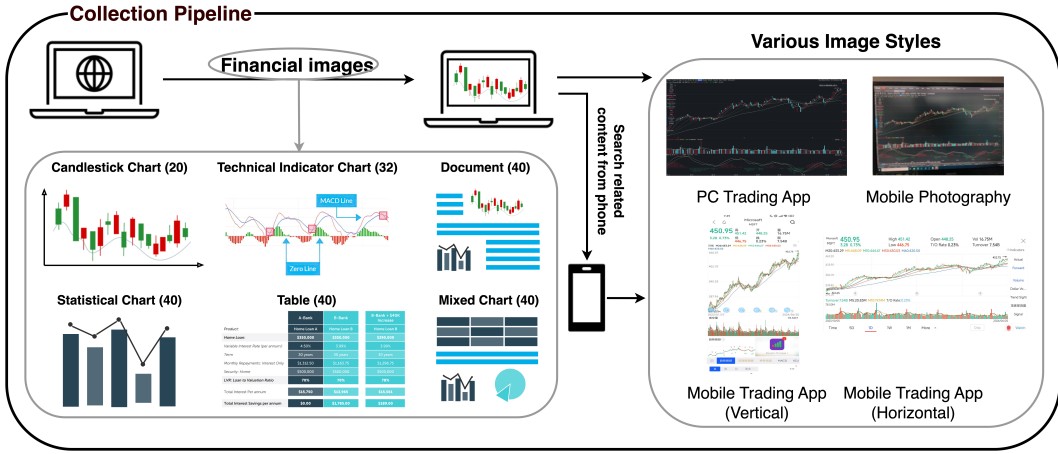

Figure 1: Data collection pipeline of MME-FINANCE.

## 3.2 DATA COLLECTION

In MME-FINANCE, we collect financial images from various mainstream platforms. Figure 1 illustrates the data collection pipeline. First, we identify relevant financial pages on a computer and use screenshot tools to capture the appropriate areas. Then, we use mobile devices to photograph the corresponding sections. Next, we search for the same content on mobile applications and capture screenshots using smartphones. The inclusion of diverse image styles, including computer screenshots, mobile photographs, and vertical and horizontal mobile screenshots, is intended to simulate real-world application scenarios. MME-FINANCE categorizes the collected images into six types: candlestick charts, technical indicator charts, statistical charts, tables, documents, and mixed charts, where a mixed chart includes at least two of other types. These images cover a broad spectrum of financial scenarios, enabling MME-FINANCE to evaluate MLLMs' ability to address challenges in this domain comprehensively.

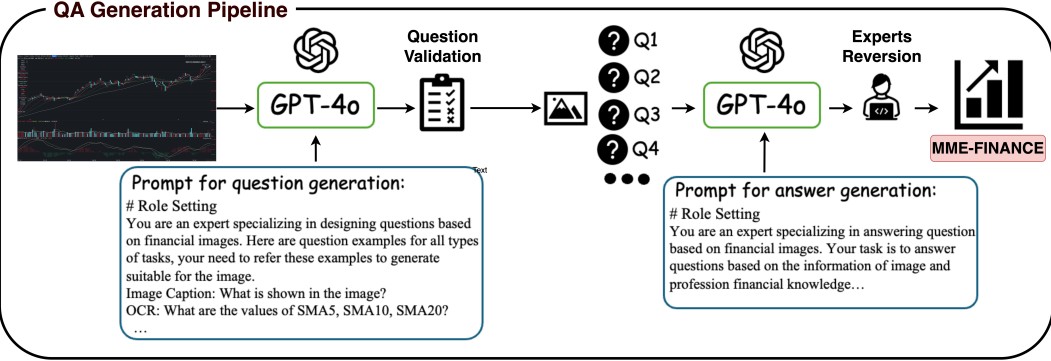

Figure 2: QA generation pipeline of MME-FINANCE.

## 3.3 QA GENERATION

To generate high-quality QA pairs for MME-FINANCE, each QA set underwent at least two stages of manual evaluation. Figure 2 illustrates the QA generation pipeline. We first design several question examples for each task. Then we utilize the GPT-4o to generate candidate questions for every image based on the example questions. We meticulously review the questions and correct inappropriate ones. In the answer generation stage, we also use GPT-4o to generate preliminary answers based on questions and images. We check all the answers manually and correct the wrong answers. The complex subjective questions are evaluated by a panel of three finance researchers, each with

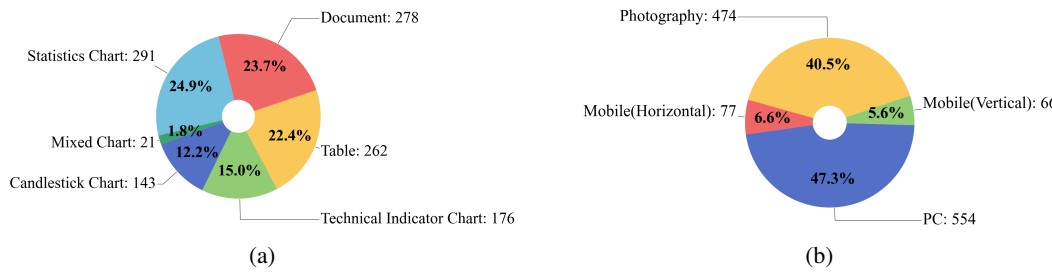

Figure 3: Distribution of different (a) types and (b) styles of images.

over 10 years of experience. The reference answer is confirmed when the reviewers reach a consensus. After this process, financial experts conduct an in-depth examination and refinement. The quality of MME-FINANCE is significantly enhanced through the manual review mechanism.

## 3.4 STATISTICS

As shown in Table 1, MME-FINANCE contains 1,171 image-question pairs spanning 11 distinct tasks, categorized into 3 ability levels as detailed in 3.1. In addition, MME-FINANCE incorporates questions aimed at evaluating hallucinations of MLLMs. The number of samples per task varies from 18 to 229, with the "Spatial Awareness" task containing the most samples and "Reason Explanation" the fewest. Figure 3(a) illustrates the distribution of the 6 image types, where statistical charts account for the main proportion, while mixed charts are the least. Figure 3(b) displays the distribution of 4 image styles. Computer screenshots and mobile photographs constitute similar proportions, representing 47.3% and 40.5% of the total, respectively. Vertical and horizontal mobile screenshots contain approximately numbers of samples.

Table 1: Statistic of the number of samples in different capabilities and tasks.

| Statistic | Number |
|---|---|
| **Perception** | 734 |
| - Image Caption | 164 |
| - OCR | 178 |
| - Entity Recognition | 163 |
| - Spatial Awareness | 229 |
| **Reasoning** | 175 |
| - Accurate Numerical Calculation | 133 |
| - Estimated Numerical Calculation | 42 |
| **Cognition** | 240 |
| - Risk Warning | 22 |
| - Investment Advice | 53 |
| - Reason Explanation | 18 |
| - Financial Question Answer | 147 |
| **Hallucination** | 22 |
| - Not Applicable | 22 |

## 3.5 EVALUATION METHOD

MME-FINANCE's question-answer format is intentionally open-ended to reflect the complexity of real-world financial scenarios. However, evaluating open-ended responses presents greater challenges compared to multiple-choice questions. To accurately evaluate the capabilities of MLLMs, we design a comprehensive evaluation process tailored to the characteristics of our benchmark. As shown in Figure 4, during the inference phase, prompts are crafted to constrain the output formats of MLLMs, thereby facilitating a more standardized evaluation. Drawing inspiration from the evaluation methodology used in MM-Vet Yu et al. (2023), we employ an Large Model (LM)-based evaluation system to compare model predictions with the ground truth and to assign a score. The scoring system is divided into six levels, ranging from 0 (completely incorrect) to 5 (fully correct), with the overall score being the average across all samples. Given the diversity in response formats across different tasks, we develop task-specific evaluation prompts to ensure accurate assessments. Additionally, a few-shot approach is employed to define scoring metrics using in-context examples, which aids the model in producing more accurate evaluation scores. Our experimental results demonstrate that the LM-based evaluator, particularly GPT-4o, achieves the highest consistency with human evaluators.

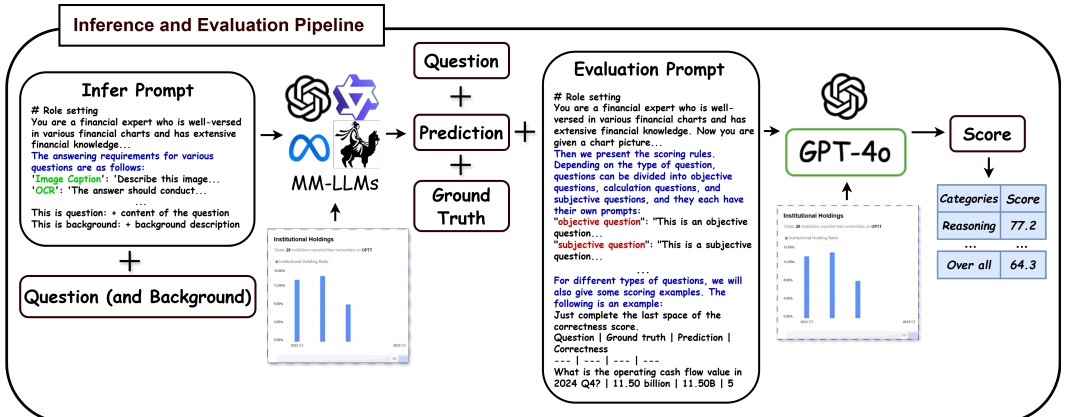

Figure 4: Inference and evaluation pipeline of MME-FINANCE. We first input the image and question prompt into the MLLMs. Then we feed the image and evaluation prompt into GPT-4o to obtain scores. The question and evaluation prompts are all designed individually for each task category.

# 4 EXPERIMENT

In this section, we introduce the experimental setup for evaluating MLLMs firstly in 4.1, followed by an exhibition and analysis of the experimental results. The main results analysis is presented in 4.2, followed by a detailed analysis of the ability dimension in 4.2.1 and image type and style dimension in 4.2.2. Finally, 4.3 elaborates on our analysis of LM as an evaluator.

## 4.1 EXPERIMENTAL SETUP

We utilize MME-FINANCE to evaluate two types of MLLMs, (1) Open-Source MLLMs including CogVLM2 (Hong et al., 2024), Qwen2-VL (team, 2024), MiniCPM-V 2.6 (Yao et al., 2024), Phi3-Vision (Abdin et al., 2024), Phi3.5-Vision (Abdin et al., 2024), LLaMA3.2 (Meta, 2024), LLaVA-NEXT (Liu et al., 2024b), YiVL (AI et al., 2024), and InternVL2 (Chen et al., 2024b); (2) Proprietary MLLMs including GPT-4o, GPT-4o mini (Open-AI, 2024). The inference prompts are the same for all MLLMs for a fair comparison, and a zero-shot setting is adopted. We fill the prompt template with image, question, ground truth, and response from an MLLM, and take the filled prompt into an LM-based evaluator for generating a score range from 0 to 5 for one sample. The scores are multiplied by 20% to be normalized.

## 4.2 MAIN RESULTS

Table 2 shows the results of various MLLMs on MME-FINANCE from the view of each task. Performance across the MLLMs varies significantly, with many models exhibiting low accuracy, highlighting the challenging nature of the MME-FINANCE benchmark. Among the evaluated models, QwenVL2-72B achieves the best overall performance with 65.69% accuracy, excelling in most tasks, particularly OCR and ANC. Proprietary MLLM, i.e., GPT-4o, ranks second overall but surpasses QwenVL2-72B in all cognition-related tasks. This suggests that GPT-4o's superior language processing capabilities give it an advantage in tasks requiring complex reasoning. Additionally, our findings support the observation from MMBench (Liu et al., 2023) that the size of the language model has a significant impact on performance. For instance, larger models in the same series, such as LLaVA-NEXT-13B compared to LLaVA-NEXT-7B, consistently demonstrate better results.

### 4.2.1 ABILITY DIMENSIONAL ANALYSIS

**Perception.** The "Perception" ability encompasses four tasks: Image Captioning (IC), Optical Character Recognition (OCR), Entity Recognition (ER), and Spatial Awareness (SA), all of which primarily focus on visual understanding. MLLMs tend to perform relatively well in the IC task, suggesting that current models exhibit satisfactory general visual perception capabilities. However, the

Table 2: Evaluation results on MME-FINANCE for all tasks. Abbreviations adopted: IC for Image Caption; ER for Entity Recognition; SA for Spatial Awareness; FQA for Financial Question Answer; ANC for Accurate Numerical Calculation; ENC for Estimated Numerical Calculation; RW for Risking Warning; IA for Investment Advice; RE for Reason Explaination; NA for Not Applicable. The first, the second, and the third highest values are highlighted by green, orange, and blue backgrounds. All numbers are denoted in % with the max value of 100%.

| Model | Overall | Perception | | | | Reasoning | | Cognition | | | | NA |
| --- | --- | --- | --- | --- | --- | --- | --- | --- | --- | --- | --- | --- |
| | | IC | OCR | ER | SA | ANC | ENC | RW | IA | RE | FQA | |
| Open source MLLMs | | | | | | | | | | | | |
| Yi-VL-34B | 17.57 | 29.39 | 1.46 | 3.93 | 8.73 | 5.56 | 11.43 | 42.73 | 35.09 | 58.89 | 47.48 | 36.36 |
| CogVLM2-19B | 40.81 | 55.24 | 51.80 | 33.13 | 14.06 | 42.11 | 27.14 | 59.09 | 52.83 | 31.11 | 50.61 | 93.64 |
| InternVL2-2B | 28.88 | 43.29 | 32.81 | 14.72 | 14.76 | 18.65 | 18.57 | 59.09 | 50.94 | 60.00 | 41.36 | 30.91 |
| InternVL2-4B | 43.71 | 58.54 | 52.92 | 31.53 | 17.29 | 52.78 | 26.19 | 68.18 | 54.34 | 64.44 | 55.10 | 59.09 |
| InternVL2-8B | 42.51 | 57.80 | 55.28 | 28.96 | 17.73 | 44.96 | 25.71 | 72.73 | 60.75 | 76.67 | 48.71 | 57.27 |
| LLaMA3.2-11B | 42.51 | 62.44 | 39.10 | 32.02 | 14.50 | 55.79 | 37.14 | 60.00 | 50.57 | 68.89 | 57.55 | 61.82 |
| LLaMA3.2-90B | 48.76 | 64.27 | 46.74 | 41.27 | 25.85 | 55.64 | 22.86 | 63.64 | 61.13 | 64.44 | 65.58 | 81.82 |
| LLaVA-NEXT-7B | 18.79 | 37.44 | 14.04 | 8.22 | 7.16 | 5.86 | 6.19 | 45.45 | 47.55 | 12.22 | 35.24 | 19.09 |
| LLaVA-NEXT-13B | 31.37 | 62.68 | 25.39 | 22.58 | 10.31 | 12.63 | 9.05 | 47.27 | 40.00 | 12.22 | 59.46 | 78.18 |
| MiniCPM2.6 | 51.65 | 71.22 | 63.71 | 37.67 | 24.37 | 55.64 | 21.43 | 72.73 | 58.87 | 66.67 | 66.80 | 77.27 |
| Phi3-Vision | 36.91 | 54.39 | 43.03 | 20.73 | 12.49 | 39.40 | 21.43 | 65.45 | 58.11 | 68.89 | 46.94 | 72.73 |
| Phi3.5-Vision | 38.99 | 67.56 | 33.03 | 18.90 | 20.52 | 32.33 | 19.52 | 67.27 | 55.85 | 72.22 | 54.42 | 93.64 |
| Qwen2VL-2B | 36.70 | 50.49 | 53.71 | 23.68 | 16.07 | 37.89 | 18.10 | 53.63 | 44.53 | 58.89 | 39.46 | 63.64 |
| Qwen2VL-7B | 36.45 | 49.76 | 50.34 | 22.58 | 15.81 | 37.44 | 21.43 | 57.27 | 48.30 | 58.89 | 42.18 | 59.09 |
| Qwen2VL-72B | 65.69 | 82.56 | 87.52 | 55.46 | 27.16 | 83.76 | 40.95 | 78.18 | 65.66 | 77.78 | 75.37 | 90.91 |
| Proprietary MLLMs | | | | | | | | | | | | |
| GPT-4o-05-13 | 42.85 | 71.34 | 28.09 | 28.22 | 19.65 | 31.73 | 36.19 | 76.36 | 62.26 | 75.56 | 71.43 | 81.82 |
| GPT-4o-mini | 57.34 | 79.15 | 68.99 | 40.25 | 24.72 | 63.31 | 43.81 | 73.64 | 64.53 | 77.78 | 73.20 | 100.0 |
| GPT-4o | 63.18 | 83.66 | 79.21 | 49.81 | 27.07 | 71.88 | 44.76 | 84.54 | 70.57 | 80.0 | 76.87 | 93.64 |

SA task proves to be the most challenging, with an highest accuracy of only 27.16%. This difficulty likely stems from the need for fine-grained perceptual abilities in the SA task. For example, as shown in Figure 5, the task requires identifying the highest Moving Average (MA) line. Several MA lines are closely positioned in the image, making it difficult for the models to distinguish between them. This suggests that while MLLMs demonstrate the competence in general visual tasks, there is still significant room for improvement in tasks requiring more precise visual discrimination.

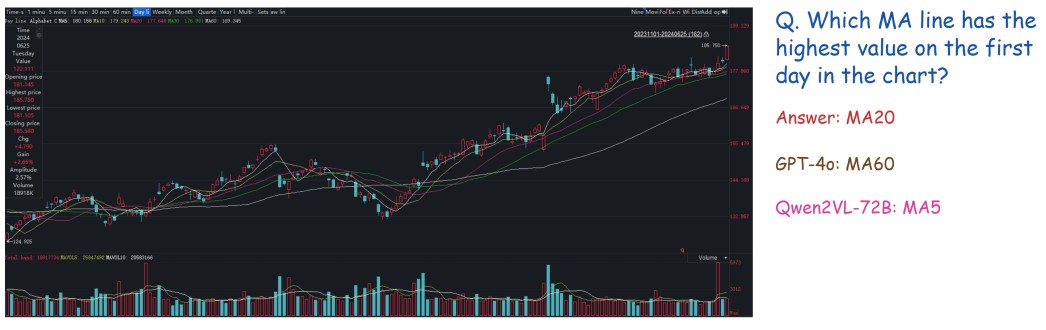

Q. Which MA line has the highest value on the first day in the chart?

Answer: MA20

GPT-4o: MA60

Qwen2VL-72B: MA5

Figure 5: Hard example in SA task of perception capability.

**Reasoning.** The "Reasoning" ability consists of two tasks: Accurate Numerical Calculation (ANC) and Estimated Numerical Calculation (ENC), both of which focus on mathematical and logical reasoning. Among these, the ENC task is significantly more challenging. This difficulty arises from the inherent complexity of estimating reasonable numerical values, a task that current MLLMs

Table 3: Evaluation results on MME-FINANCE for different types and formats of images. Abbreviations adopted: Candle. for Candlestick chart; Tech. for Technical indicator chart; Stat. for Statistical chart; Tab. for Table; Doc. for Document; Mixed for Mixed chart; CS for Computer Screenshot; MP for Mobile Photograph; VS for Vertical Screenshot on Mobile; HS for Horizontal Screenshot on Mobile. The first, the second, and the third highest values are highlighted by green , orange , and blue backgrounds. All numbers are denoted in % with the max value of 100%.

| Model | Candle. | Tech. | Stat. | Tab. | Doc. | Mixed | CS | MP | VS | HS |
|---|---|---|---|---|---|---|---|---|---|---|
| Open source MLLMs | | | | | | | | | | |
| Yi-VL-34B | 23.64 | 16.36 | 18.76 | 15.42 | 14.89 | 32.38 | 19.42 | 14.39 | 26.06 | 16.62 |
| CogVLM2-19B | 5.03 | 26.93 | 52.30 | 50.38 | 45.76 | 57.14 | 44.91 | 41.98 | 17.58 | 24.16 |
| InternVL2-2B | 23.78 | 4.55 | 38.62 | 24.65 | 38.49 | 58.10 | 33.43 | 28.57 | 14.24 | 10.65 |
| InternVL2-4B | 5.45 | 36.82 | 51.48 | 54.66 | 47.77 | 63.81 | 48.77 | 42.91 | 23.33 | 29.61 |
| InternVL2-8B | 42.38 | 45.00 | 25.98 | 57.79 | 42.01 | 67.62 | 46.93 | 35.27 | 48.79 | 49.87 |
| LLaMA3.2-11B | 35.24 | 31.59 | 47.63 | 50.92 | 39.42 | 48.57 | 45.16 | 39.07 | 38.79 | 47.79 |
| LLaMA3.2-90B | 40.56 | 40.11 | 51.20 | 58.17 | 45.83 | 64.76 | 50.14 | 46.33 | 46.06 | 56.10 |
| LLaVA-NEXT-7B | 29.65 | 23.52 | 28.80 | 15.65 | 0.72 | 44.76 | 17.87 | 15.23 | 32.73 | 35.32 |
| LLaVA-NEXT-13B | 27.27 | 26.36 | 33.68 | 32.14 | 32.95 | 39.05 | 32.67 | 29.20 | 30.91 | 35.84 |
| MiniCPM2.6 | 45.03 | 45.00 | 54.23 | 58.63 | 49.42 | 59.05 | 52.09 | 50.51 | 45.45 | 60.78 |
| Phi3-Vision | 37.62 | 40.00 | 42.06 | 49.54 | 15.32 | 62.86 | 41.59 | 28.48 | 40.30 | 52.21 |
| Phi3.5-Vision | 32.73 | 30.45 | 46.25 | 38.24 | 39.21 | 59.05 | 44.73 | 32.28 | 41.52 | 36.88 |
| Qwen2VL-2B | 6.43 | 15.68 | 46.60 | 46.26 | 44.68 | 57.14 | 41.05 | 38.95 | 8.18 | 16.10 |
| Qwen2VL-7B | 12.03 | 11.02 | 46.60 | 46.11 | 44.03 | 54.29 | 40.58 | 39.41 | 6.67 | 14.03 |
| Qwen2VL-72B | 60.12 | 60.11 | 65.15 | 71.73 | 66.04 | 74.24 | 67.65 | 62.78 | 68.48 | 67.01 |
| Proprietary MLLMs | | | | | | | | | | |
| GPT-4o-05-13 | 44.62 | 32.84 | 52.99 | 37.18 | 41.65 | 60.95 | 46.43 | 37.26 | 47.27 | 47.79 |
| GPT-4o-mini | 51.89 | 50.91 | 63.37 | 60.46 | 54.10 | 68.57 | 59.71 | 53.45 | 62.12 | 60.0 |
| GPT-4o | 58.32 | 55.68 | 67.84 | 68.55 | 59.71 | 73.33 | 65.67 | 58.31 | 69.09 | 70.13 |

struggle with. As shown in Table 2, the best-performing model, i.e., Qwen2VL-72B, achieved only 40.95% accuracy on the ENC task, which is much lower compared to 83.76% on the ANC task. This discrepancy highlights the continued difficulty MLLMs face in handling estimation-based reasoning problems. As depicted in Figure 6, the exact numerical values are not explicitly presented in the image, requiring the model to infer these values based on contextual clues, such as spatial relationships. The inability to reasonably estimate such values remains a critical limitation of current MLLMs.

**Cognition.** The "Cognition" task, consisting of Risk Warning (RW), Investment Advice (IA), Reason Explanation (RE), and Financial Knowledge QA (FQA), assesses the ability of MLLMs to make complex financial decisions. Due to the inherently subjective nature of these questions, the performance variance among different models is smaller compared to other tasks. This suggests that current MLLMs demonstrate a basic competence in financial reasoning. GPT-4o achieves the highest overall score across those 4 tasks, indicating its superior capability in handling financially complex and subjective decision-making scenarios.

**Hallucination Problem.** The Not Applicable (NA) task is specifically designed to assess the hallucination of MLLMs. For this task, we developed an inference prompt that explicitly informs the models that they can respond with "Not Applicable" if they determine that no suitable answer is attainable. As seen in Table 2, models such as CogVLM2-19B, Phi3.5-Vision, Qwen2VL-72B, and GPT-4o demonstrate a strong ability to discern whether a question is answerable, thereby mitigating the hallucination issue. In contrast, models like Yi-VL-34B, InternVL2-2B, and LLaVA-NEXT-7B exhibit severe hallucination problems. Since the prompt reminds MLLMs of the "Not Applicable" option in the NA task, the task difficulty is somewhat reduced. To explore hallucination issues more comprehensively, we modified the prompt to allow "Not Applicable" response across all types of tasks. This led to a rise in false negatives in most MLLMs, where models incorrectly marked

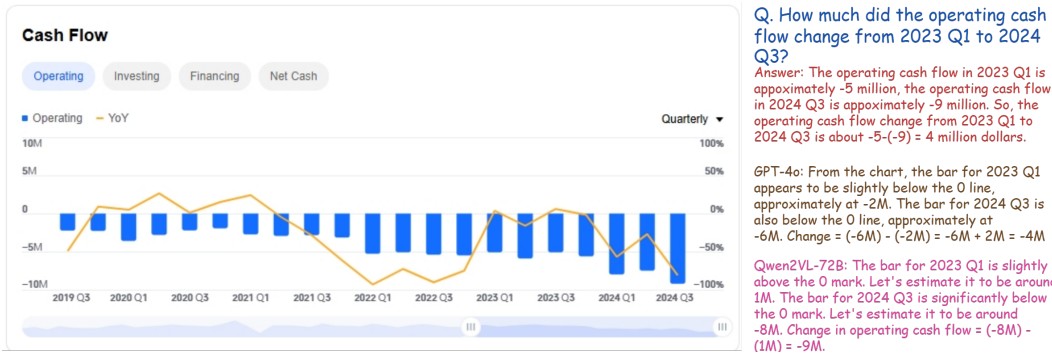

Figure 6: Hard example in ENC task of reasoning capability.

questions with valid answers as unanswerable, suggesting that hallucination remains a significant challenge for many MLLMs. The corresponding experimental results are detailed in the appendix.

### 4.2.2 IMAGE TYPE AND STYLE DIMENSIONAL ANALYSIS

Table 3 presents the performance of various MLLMs from the perspective of image types and styles. Notably, most models exhibit poor performance on candlestick charts and technical indicator charts. This can be attributed to the specialized nature of these charts, which demand domain-specific knowledge that current MLLMs struggle to interpret. Regarding image styles, MLLMs perform particularly poorly on mobile photographs, mainly due to the lower resolution typical of phone-captured images, which obscures critical details. Furthermore, many of these photos are taken at oblique angles, leading to incomplete or extraneous visual information. Considering that such images are prevalent in real-world applications, it is crucial to improve MLLMs' ability so as to effectively process and analyze mobile photographs.

### 4.3 ANALYSIS OF EVALUATORS

For a fair comparison of the tested MLLMs, we conduct extensive experiments to assess the effectiveness of various evaluators. First, we selected 100 samples and generated corresponding outputs from MiniCPM2.6. To ensure objectivity, each sample is scored by three experienced experts. The final score for each sample is determined by mode (e.g., 3 for scores of 2, 3, and 3) or mean (e.g., 2 for scores of 1, 2, and 3). If the score variance exceeds 2, the sample is subjected to further review to assign a final score. Then, Spearman's rank correlation coefficient is calculated to indicate the evaluators' performance. As shown in Table 4, GPT-4o with image input achieves the highest Spearman's rank correlation coefficient and the lowest average absolute difference, indicating superior performance. GPT-4-Turbo also demonstrates strong performance, significantly outperforming GPT-3.5-Turbo and o1-preview. Among the open-source evaluators, Qwen72B achieves the best results, substantially surpassing CogVLM2 and MiniCPM2.6. Furthermore, we observed that most evaluators perform better with additional image inputs. We believe this improvement is due to the images that provide evaluators with additional information. We further divide the questions into two categories: subjective and objective. From Table 5, it can be seen that the GPT-4o evaluator exhibits higher consistency in scoring objective questions, and the inclusion of images notably improves accuracy in evaluating subjective questions.

To the best of our knowledge, images have never been included as part of the input to the evaluator in previous work. Our findings may contribute to refining evaluation methodologies for MLLMs and encourage further exploration within the research community.

## 5 CONCLUSION

In this paper, we introduce MME-FINANCE, a pioneering effort to establish a comprehensive multi-modal benchmark tailored to evaluate the capabilities of MLLMs within the specialized financial domain. To the best of our knowledge, MME-FINANCE is the first benchmark to systematically assess

Table 4: The comparison of Spearman's rank correlation coefficient (Sp.) and average absolute differences (Δ) between the evaluation scores of various evaluators and human-annotated scores. Larger Sp. and smaller Δ represent a better agreement with human evaluation, indicating a better evaluator. Abbreviations adopted: Cog. for CogVLM2-19B; Mini. for MiniCPM2.6; Qwen72B for QwenVL2-72B. Pic. represents adding image as input when evaluating.

| Model | GPT-3.5Turbo | GPT-4Turbo | o1-preview | GPT-4o(Pic.) | Cog.(Pic.) | Mini.(Pic.) | Qwen72B(Pic.) |
|---|---|---|---|---|---|---|---|
| Sp. (↑) | 0.498 | 0.711 | 0.592 | 0.720(0.738) | 0.049(0.027) | 0.048(0.162) | 0.688(0.678) |
| Δ (↓) | 1.39 | 0.93 | 1.14 | 0.90(0.84) | 2.18(2.27) | 2.07(1.88) | 1.02(1.01) |

Table 5: The Spearman's rank correlation coefficient (Sp.) and average absolute differences (Δ) between the evaluation scores of GPT-4o and human-annotated scores. Larger Sp. and smaller Δ represent a better agreement with human evaluation, indicating a better evaluator. Objective and Subjective denote objective and subjective questions. w and w/o represent evaluating with and without image.

| Objective | | | | Subjective | | | |
|---|---|---|---|---|---|---|---|
| w | | w/o | | w | | w/o | |
| Sp. | Δ | Sp. | Δ | Sp. | Δ | Sp. | Δ |
| 0.835 | 0.57 | 0.849 | 0.60 | 0.515 | 1.23 | 0.471 | 1.3 |

the performance of MLLMs on tasks involving multimodal, open-ended financial knowledge tasks. By encompassing a diverse range of financial images and open-ended questions, MME-FINANCE challenges models to go beyond basic visual-text alignment and instead engage in complex financial reasoning and expert-level decision-making. Moreover, a novel evaluation strategy is proposed for an accurate evaluation of the MLLMs. Our detailed evaluation of 18 MLLMs shows that both open-source and proprietary models have significant limitations in processing and reasoning about complex financial questions. MME-FINANCE can serve as a critical tool to guide the development of MLLM capabilities in the financial domain. In future work, we plan to expand MME-FINANCE by integrating multi-turn dialogue scenarios to enhance realistic human-AI collaboration in financial decision-making.

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

## A  APPENDIX

Statement: This paper is limited to academic research, and the OpenAI's products used do not violate the company's commercial regulations.

In this appendix, we provide further details regarding the proposed MME-FINANCE. A.1 presents experimental results with the prompt to allow "Not Applicable" response across all types of tasks. A.2 provides some samples demonstrating the difficulty of recognizing mobile photos and the hallucination problems of MLLMs. A.3 includes the detailed inference and evaluation prompts. A.4 exhibits example for each task.

### A.1  RESULTS OF NA PROMPT FOR ALL TASKS.

Table 6 and Table 7 shows the performance of MLLMs with the prompt to allow "Not Applicable" response across all types of tasks. It is clear that most models have a lower performance in the setting, which means the hallucination problem is quite common in MLLMs. Although some models have a high recall of the "Not Applicable" question, their overall accuracy is low. It shows that these models tend to answer "Not Applicable" for some unsure questions.

### A.2  HARD EXAMPLES

In this section, we present some hard examples about the difficulty of mobile photos and hallucination problems of MLLMs. As shown in Figure 7, the two questions have similar content. When feeding the two images into the same model, the responses are different. For the picture taken with a mobile phone, the model mistakenly identifies decimal points as commas and the letter B as the number 8. And the model accurately identifies corresponding elements in the computer screenshot. This indicates that the perception of mobile phone photos is a challenge for some MLLMs. Figure 8 illustrates a example of the hallucination problem. GPT-4o cannot recognize the initial increase trend, while Qwen2VL-72B totally unable to perceive trends.

Table 6: Evaluation results on MME-FINANCE for all tasks. Abbreviations adopted: IC for Image Caption; ER for Entity Recognition; SA for Spatial Awareness; FQA for Financial Question Answer; ANC for Accurate Numerical Calculation; ENC for Estimated Numerical Calculation; RW for Risking Warning; IA for Investment Advice; RE for Reason Explaination; NA for Not Applicable. The first, the second, and the third highest values are highlighted by green , orange , and blue backgrounds. All numbers are denoted in % with the max value of 100%.

| Model | Overall | Perception | | | | Reasoning | | Cognition | | | | NA |
|---|---|---|---|---|---|---|---|---|---|---|---|---|
| | | IC | OCR | ER | SA | ANC | ENC | RW | IA | RE | FQA | |
| Open source MLLMs | | | | | | | | | | | | |
| CogVLM2-19B | 31.24 | 36.22 | 42.02 | 26.99 | 7.95 | 34.14 | 19.52 | 13.64 | 38.11 | 32.22 | 48.44 | 70.91 |
| InternVL2-2B | 32.16 | 61.22 | 32.81 | 12.76 | 4.72 | 32.78 | 21.43 | 54.55 | 47.92 | 63.33 | 50.20 | 50.00 |
| InternVL2-4B | 45.93 | 68.05 | 54.27 | 28.22 | 18.69 | 54.74 | 32.38 | 68.18 | 50.19 | 68.89 | 59.46 | 59.09 |
| InternVL2-8B | 50.59 | 70.00 | 60.11 | 33.99 | 23.84 | 62.56 | 40.00 | 76.36 | 60.75 | 75.56 | 58.37 | 55.45 |
| LLaVA-NEXT-7B | 20.10 | 58.05 | 6.18 | 2.70 | 1.31 | 6.62 | 3.33 | 43.64 | 37.74 | 16.67 | 43.27 | 70.00 |
| MiniCPM2.6 | 48.37 | 69.63 | 62.81 | 34.85 | 21.31 | 54.89 | 28.57 | 50.00 | 40.38 | 45.56 | 62.31 | 80.00 |
| Phi3-Vision | 37.06 | 69.51 | 58.43 | 29.57 | 11.88 | 22.11 | 3.81 | 7.27 | 16.60 | 41.11 | 47.76 | 98.18 |
| Phi3.5-Vision | 28.69 | 66.83 | 33.03 | 13.87 | 8.12 | 17.14 | 6.19 | 2.73 | 15.47 | 22.22 | 45.58 | 96.36 |
| Qwen2VL-7B | 32.40 | 62.32 | 40.00 | 8.10 | 7.07 | 41.80 | 22.86 | 33.64 | 40.0 | 10.0 | 40.14 | 100.00 |
| Qwen2VL-2B | 32.47 | 62.32 | 40.45 | 9.08 | 6.72 | 41.80 | 25.24 | 31.82 | 36.23 | 13.33 | 40.14 | 100.00 |
| QwenVL2-72B | 62.97 | 80.49 | 83.26 | 50.43 | 25.50 | 78.95 | 46.67 | 73.64 | 68.30 | 73.33 | 73.06 | 86.36 |
| Proprietary MLLMs | | | | | | | | | | | | |
| GPT-4o-5-13 | 42.12 | 72.07 | 26.74 | 26.99 | 19.56 | 29.17 | 38.57 | 70.90 | 63.77 | 76.67 | 71.02 | 72.72 |
| GPT-4o-mini | 41.32 | 63.54 | 58.54 | 28.34 | 14.32 | 24.06 | 6.19 | 52.73 | 30.19 | 63.33 | 68.57 | 100.00 |
| GPT-4o | 61.35 | 83.66 | 78.54 | 46.38 | 28.73 | 71.43 | 40.00 | 80.00 | 65.66 | 77.78 | 70.88 | 80.00 |

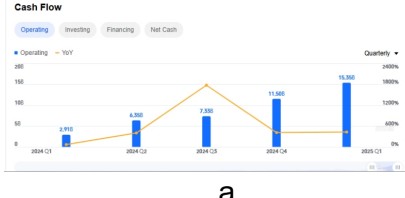

Q. How much did the cash flow increase from 2024 Q3 to 2025 Q1?

Answer: The cash flow in 2024 Q3 is 7.33B, and the cash flow in 2025 Q1 is 15.35, hence the increase is 15.35-7.33=8.02B.

a

Prediction(a): The cash flow for 2024 Q3 is 7.33B, and for 2025 Q1, it is 15.35B. The calculation is as follows: 15.35B - 7.33B = 8.02B.

Prediction(b): The cash flow for 2024 Q3 is 7,338 and for 2025 Q1 is 15,358. The calculation is as follows: 15,358 - 7,338 = 8,020.

b

Figure 7: Comparison of the difficulty of recognizing computer screenshot versus photos taken with a mobile phone.

A.3  INFERENCE AND EVALUATION PROMPT.

Figure 9 and Figure 10 shows the detailed inference prompt and evaluation prompt.

A.4  EXAMPLES FOR EACH TASK.

Table 7: Evaluation results on MME-FINANCE for different types and formats of images. Abbreviations adopted: Candle. for Candlestick chart; Tech. for Technical indicator chart; Stat. for Statistical chart; Tab. for Table; Doc. for Document; Mixed for Mixed chart; CS for Computer Screenshot; MP for Mobile Photograph; VS for Vertical Screenshot on Mobile; HS for Horizontal Screenshot on Mobile. The first, the second, and the third highest values are highlighted by green, orange, and blue backgrounds. All numbers are denoted in % with the max value of 100%.

| Model | Candle. | Tech. | Stat. | Tab. | Doc. | Mixed | CS | MP | VS | HS |
|---|---|---|---|---|---|---|---|---|---|---|
| Open source MLLMs | | | | | | | | | | |
| CogVLM2-19B | 25.31 | 19.89 | 33.13 | 32.52 | 37.53 | 39.05 | 31.70 | 29.66 | 33.33 | 35.84 |
| InternVL2-2B | 25.03 | 25.57 | 36.64 | 32.37 | 33.88 | 53.33 | 36.21 | 28.82 | 25.76 | 29.09 |
| InternVL2-4B | 30.91 | 34.43 | 54.35 | 45.47 | 52.37 | 56.19 | 49.68 | 43.63 | 39.39 | 38.70 |
| InternVL2-8B | 39.02 | 38.18 | 55.27 | 50.72 | 58.08 | 69.52 | 54.98 | 47.13 | 49.70 | 41.04 |
| LLaVA-NEXT-7B | 16.08 | 18.75 | 22.82 | 19.31 | 19.93 | 33.33 | 22.42 | 16.71 | 22.12 | 22.60 |
| MiniCPM2.6 | 39.86 | 44.66 | 51.53 | 48.85 | 52.23 | 38.10 | 47.26 | 48.48 | 48.48 | 55.58 |
| Phi3-Vision | 24.62 | 28.18 | 42.82 | 40.00 | 42.16 | 12.38 | 38.66 | 35.86 | 29.70 | 39.22 |
| Phi3.5-Vision | 20.98 | 22.95 | 30.23 | 30.29 | 34.36 | 10.48 | 32.92 | 22.95 | 31.82 | 30.91 |
| Qwen2VL-7B | 22.80 | 24.32 | 35.34 | 37.63 | 33.20 | 48.57 | 33.86 | 30.42 | 29.39 | 36.62 |
| Qwen2VL-2B | 23.92 | 23.86 | 35.57 | 37.91 | 33.06 | 43.81 | 33.54 | 31.18 | 29.39 | 35.32 |
| Qwen2VL-72B | 52.31 | 56.14 | 69.16 | 64.32 | 64.67 | 74.29 | 65.78 | 59.41 | 63.94 | 63.90 |
| Proprietary MLLMs | | | | | | | | | | |
| GPT-4o-5-13 | 39.86 | 33.86 | 53.13 | 36.72 | 40.43 | 63.81 | 45.67 | 36.58 | 49.39 | 44.41 |
| GPT-4o-mini | 26.15 | 26.02 | 49.90 | 46.87 | 43.88 | 50.48 | 45.99 | 36.50 | 35.76 | 42.08 |
| GPT-4o | 52.45 | 51.93 | 67.70 | 68.79 | 57.12 | 76.19 | 64.77 | 55.82 | 66.67 | 66.23 |

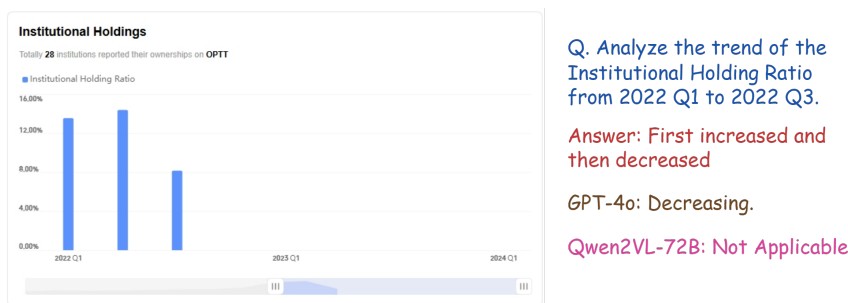

Figure 8: The display of hallucination problems of MLLMs.

**STEP1**: We provide a general prompt: 'You are a financial expert who is well - versed in various financial charts and has extensive financial knowledge. Now you are given an image and a corresponding question. Please answer this question.'

**STEP2**: We present the question: 'This is question: + content of the question'

**STEP3**: We state the requirements for answering the question. Here, for different types of questions, the answering requirements are also different: 'Here are the answer requirements: + answering requirements'

The answering requirements for various questions are as follows:

'Image Caption': 'Describe this image in a whole - part structure. Start with a sentence summarizing the main theme of the image. If the image depicts multiple objects, first introduce each object in one sentence, and if there are some connections between objects, explain each connection in one sentence. If the object is complex, it can be further explained. Your answer should be less than 250 words and should not include any irrelevant information.'

'OCR': 'The answer should conduct an Optical Character Recognition (OCR) analysis on the content inquired about. Just answer the question with a single word or phrase if possible. No irrelevant information should be included.'

'Entity Recognition': 'The answer should contain recognition results of entities mentioned in the question. Just answer the question with a single word or phrase if possible. No irrelevant information should be included.'

'Spatial Awareness': 'The answer should be based on the spatial relationships between entities in the question. It is best to provide corresponding evidence for all judgments. If specific numerical answers are not present in the image but can be estimated based on its content, the estimated results can be used. Just provide the answer in one word or a short sentence. No irrelevant information should be included.'

'Numerical Calculation': 'You should perform mathematical calculations based on the information in the image. You need to estimate some values that are not directly displayed in the image for answering the question. You should show the calculation process and output the calculated result.'

'Accurate Numerical Calculation': 'You should perform mathematical calculations based on the information in the image. You need to provide a step - by - step calculation and obtain a numerical result.'

'Financial Knowledge': 'The answer should be based on financial knowledge. Briefly answer the question within 100 words. The answer should not contain irrelevant content related to the picture.'

'Risk Warning': 'You should warn of investment risk based on the information in the chart and professional financial knowledge. All your arguments need to be supported by facts or theories and the answer should be within 150 words.'

'Investment Advice': 'You should provide investment advice based on the information in the chart and professional financial knowledge. All your arguments need to be supported by facts or theories and the answer should be within 150 words.'

'Explain Reason': 'You should provide an explanation based on the information in the chart and professional financial knowledge. All your arguments need to be supported by facts or theories and the answer should be within 150 words.'

'Not Applicable': 'If you cannot answer, please say "Not Applicable", and provide the explanations.'

**STEP4**: If the question has background information, we will add a background description for it: 'This is background: + background description'

Figure 9: Inference prompt.

STEP1: We provide a general prompt: "You are a financial expert who is well - versed in various financial charts and has extensive financial knowledge. Now you are given an image and a corresponding question. Please answer this question."

STEP2: Then we present the scoring rules. Depending on the type of question, questions can be divided into objective questions, accurate calculation questions, numerical calculation questions, subjective questions, and not Applicable question, and they each have their own prompts, as shown below:
"objective question": "This is an objective question. Please give a score: (The full score is 5 points in total. Score according to the following conditions.)
Answer accuracy: Full score is 5 points. In combination with the question, it is required that the content and semantics of the prediction and the answer must be the same and there should be no redundant answers. The answer can be expressed in different ways, such as different unit symbols and different counting methods. If the answer is correct, 5 points can be given. If the answer contains multiple pieces of content, multiply 5 by the correct proportion of the prediction to give the final score. If the answer is wrong, give 0 points directly."
"accurate calculation question": "This is a calculation question. Please give a score: (The full score is 5 points in total. Score item by item according to the following conditions and add up the obtained scores to get the total score.)
1. Answer accuracy: Full score is 2 points. In combination with the question and answer, it is required that the final calculated result of the prediction must be accurate. If the answer is correct, give 2 points. If the answer is wrong, give 0 points.
2. Calculation process: Full score is 3 points. There should be intermediate calculation processes for calculation questions, and they should also be correct. In combination with the answer, if all elements and steps are included in the prediction, give 3 points. If the final answer is wrong but the calculation process included in the prediction is partially correct, multiply 3 by the correct proportion to give the final score. If the calculation process is also wrong, give 0 points."
"numerical calculation question": "This is a valuation calculation question. Please give a score: (The full score is 5 points in total. Score item by item according to the following conditions and add up the obtained scores to get the total score.)
1. Answer accuracy: Full score is 2 points. In combination with the question and answer, if the predicted final result fluctuates within ±10% of the final value of the answer, give 2 points. If the predicted final result fluctuates within ±10% - ±20% of the final value of the answer, give 1 point. If the predicted final result fluctuates more than ±20% of the final value of the answer, give 0 points.
2. Calculation process: Full score is 3 points. There should be intermediate calculation processes for calculation questions, and they should also be correct. In combination with the answer, if all elements and steps are included in the prediction, give 3 points. If the final answer is wrong but the calculation process included in the prediction is partially correct, multiply 3 by the correct proportion to give the final score. If the calculation process is also wrong, give 0 points."
"subjective question": "This is a subjective question. Please give a score: (The full score is 5 points in total. Score item by item according to the following conditions and add up the obtained scores to output the total score.)
1. Content matching degree: Full score is 2 points. When all keywords of the answer appear in the predicted text, give 2 points. When some keywords of the answer appear in the predicted text, give 1 point. When none of the keywords of the answer appear in the predicted text, give 0 points.
2. Semantic matching degree: Full score is 2 points. When the semantics of the answer and the predicted content are close and there is no wrong judgment, give 2 points. When part of the semantics of the answer and the predicted content are close, give 1 point. When the semantics of the answer and the predicted content are completely different, give 0 points.
3. Problem attribute self-consistency: Full score is 1 point. Give points as appropriate according to the following prediction requirements, and require smooth logic and correct grammar."
"not Applicable question": "This is an unanswerable question. Please give a score: (The full score is 5 points in total. Score according to the following conditions.)
Answer accuracy: Full score is 5 points. If 'Not Applicable' appears in the prediction result, give 5 points directly. If 'Not Applicable' does not appear in the prediction but indicates that it cannot be answered, give points as appropriate."

At the same time, we will also add the response requirements during inference as prompts (refer to the inference response requirements).

STEP3: For different types of questions, we will also give some scoring examples. This can provide some few - shot for the scoring model. The following is an example of 'OCR':
"Just complete the last space of the correctness score.
Question | Ground truth | Prediction | Correctness
--- | --- | --- | ---
What is the operating cash flow value in 2024 Q4? | 11.50 billion | 11.50B | 5
What are the last candlestick values for SMA 5/10/20/30/60? | The last candlestick values for the Simple Moving Averages (SMA) of periods 5, 10, 20, 30, and 60 for Microsoft Corp. (MSFT) are as follows: - SMA 5: 448.09 - SMA 10: 444.67 - SMA 20: 433.29 - SMA 30: 430.03 - SMA 60: 420.50 | 5:448.09, 10:444.67, 30:430.03, 60:420.50 | 4 "

STEP4: Finally, we will arrange the questions, answers, and predictions in the same format as the above examples for the scoring model to score.The following is an example of 'OCR' question:
"What is the Turnover shown in the chart? | 3.48B | 3.48B |   "

Figure 10: Evaluation prompt.

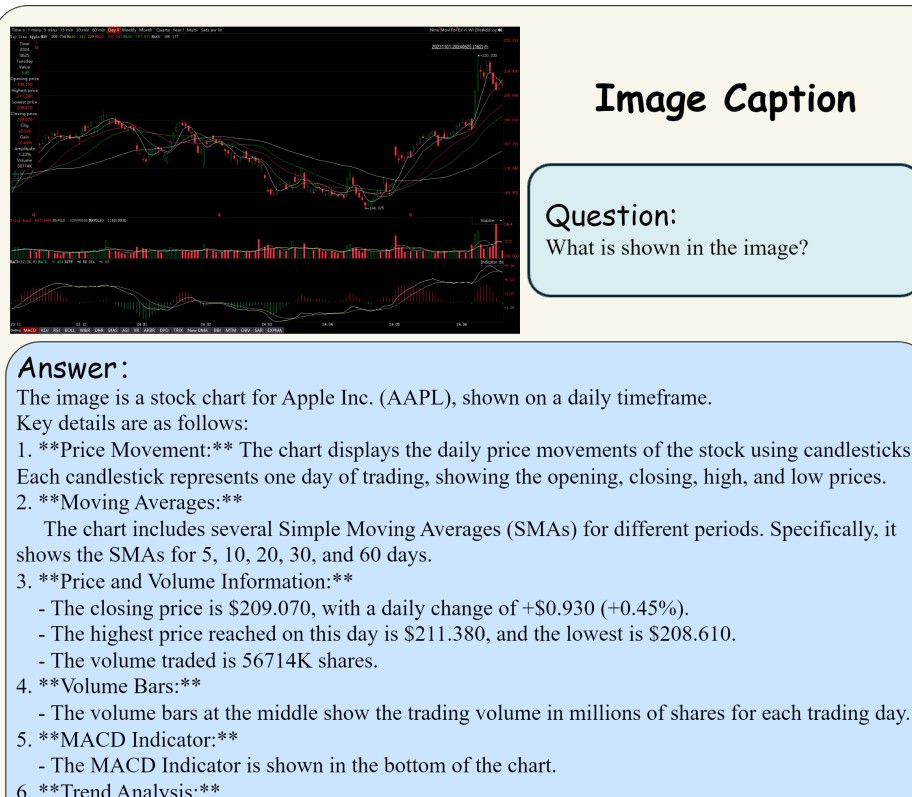

Figure 11: Image Caption.

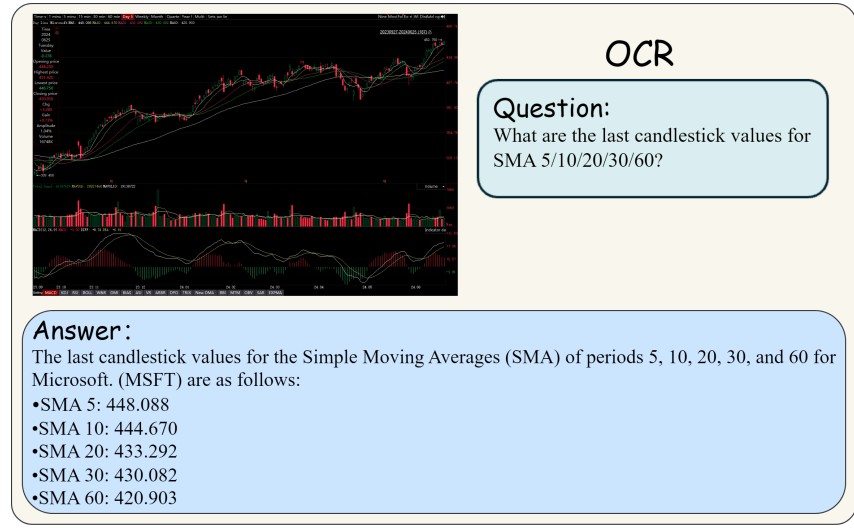

Figure 12: OCR.

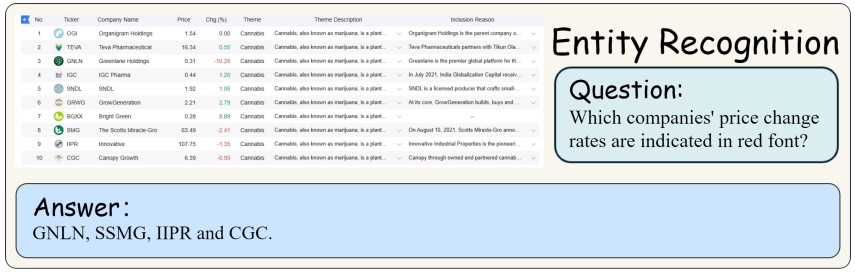

Figure 13: Entity Recognition.

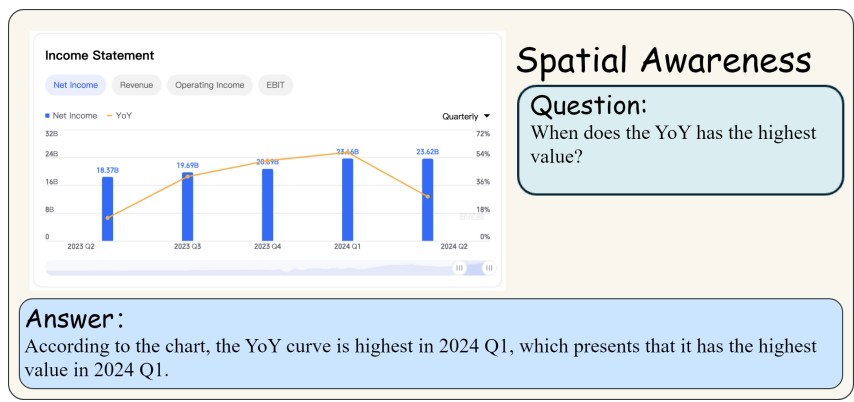

Figure 14: Spatial Awareness.

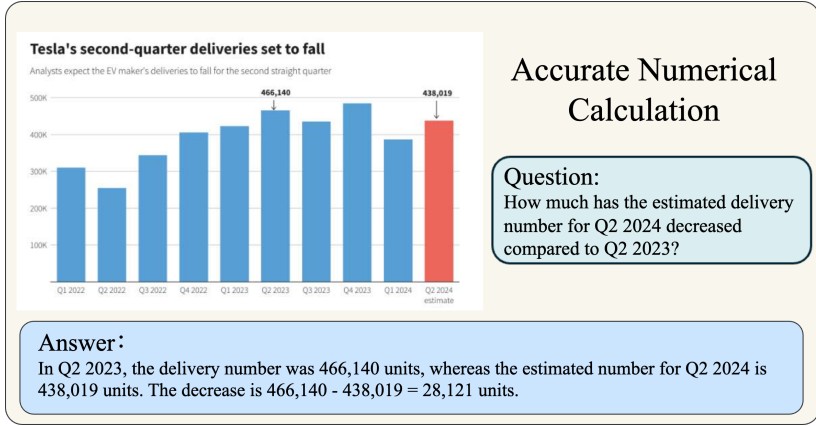

Figure 15: Accurate Numerical Calculation.

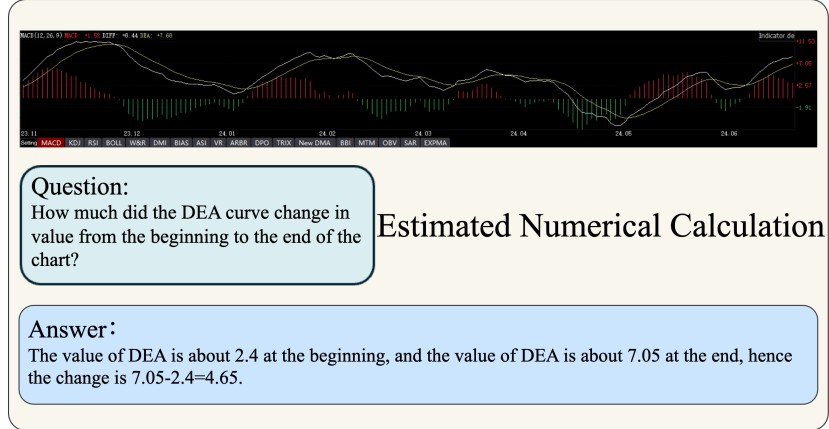

Figure 16: Estimated Numerical Calculation.

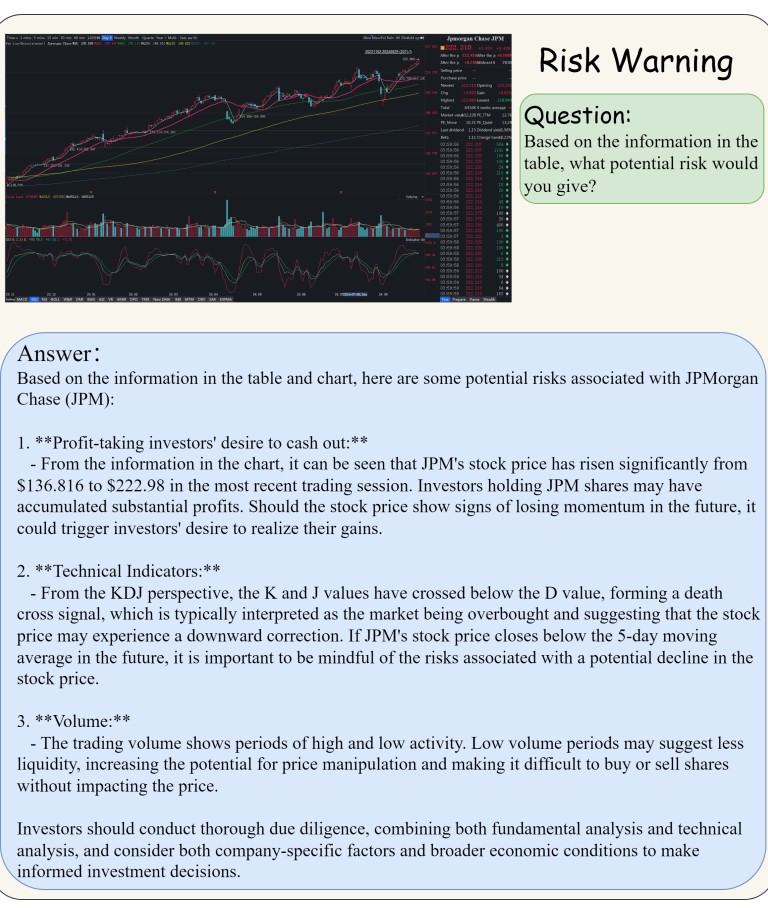

Figure 17: Risk Warning.

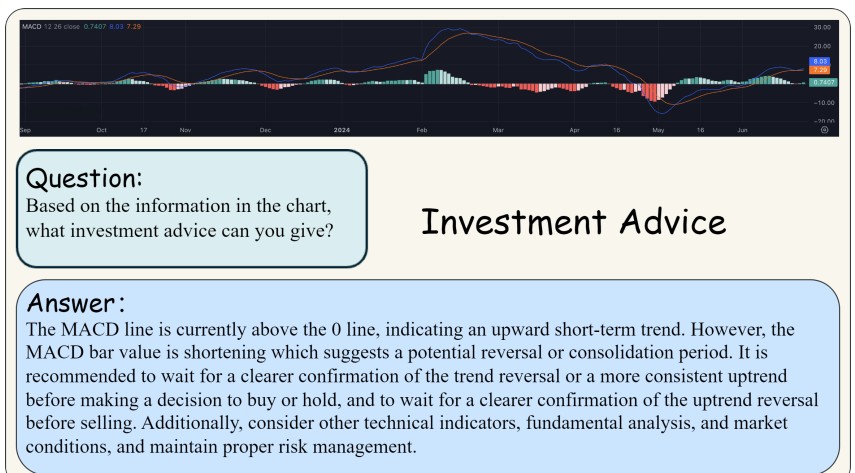

Figure 18: Investment Advice.

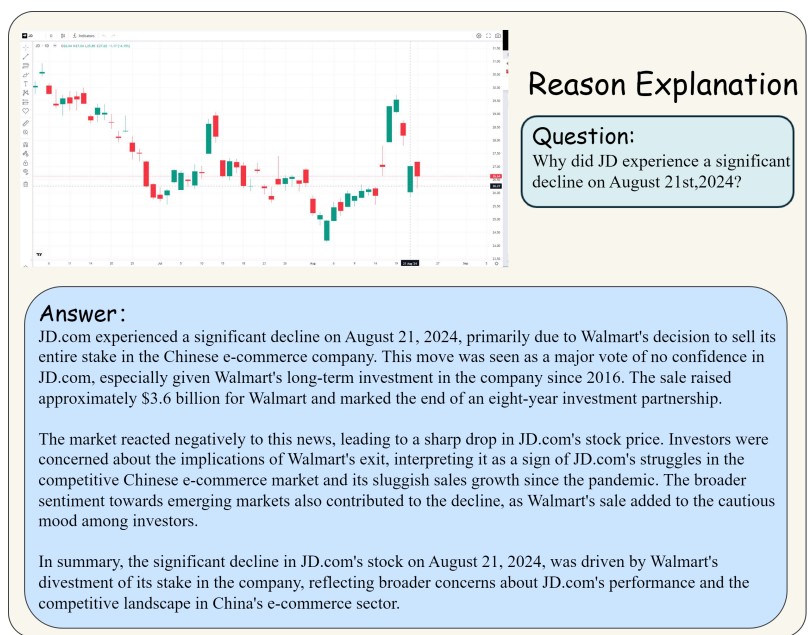

Figure 19: Reason Explanation.

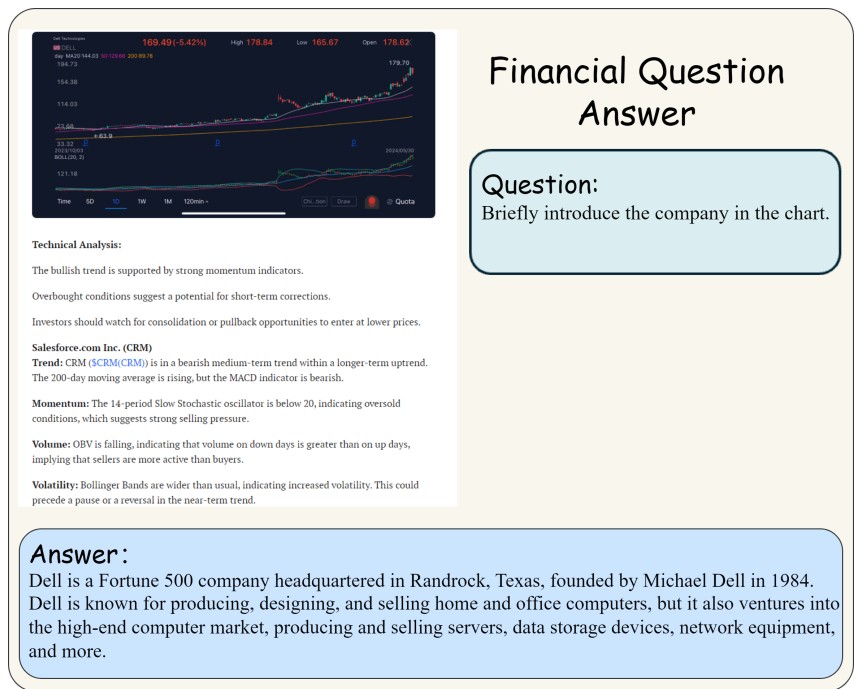

Figure 20: Financial Question Answer.

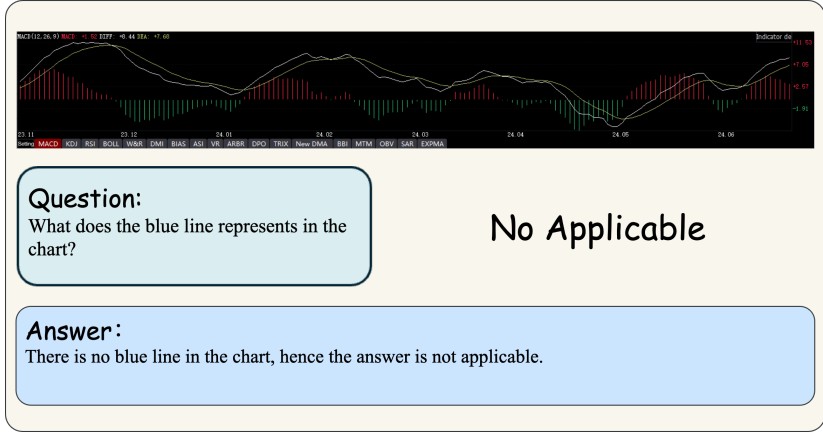

Figure 21: Not Applicable.

