# OpenReview forum: "MME-FINANCE: A Multimodal Finance Benchmark for Expert-level Understanding and Reasoning"
_ICLR.cc/2025/Conference — ICLR 2025 Conference Withdrawn Submission_

### Official Review · Reviewer_GWAX · 2024-10-28

**Soundness:** 3
**Presentation:** 2
**Contribution:** 3
**Rating:** 5
**Confidence:** 4

**Summary:**

The authors present a multimodal financial chart understanding benchmark and evaluate the performance of 18 models using a prompt-based evaluation method.

**Strengths:**

1. The first multimodal benchmark for financial knowledge Q&A, filling a gap in the field.
2. Charts are sourced from real-world data, and the Q&A has been manually reviewed and refined.
3. The models tested are fairly new and comprehensive.
4. The analysis of different MLLMs as judges is notable for evaluations.

**Weaknesses:**

1. The authors do not clarify the source of the charts, raising concerns about potential privacy or copyright issues.
2. There is a lack of overall dataset analysis. For example, is there a domain gap between the charts and other datasets like MME or SEED? If the charts come from a narrow range of sources, can the authors prove the diversity of chart styles (not just chart types) through clustering methods?
3. The GPT-4 prompt-based approach is already widely adopted, and the evaluation manner is costly.
4. The metadata only includes charts, not the ground truth (GT) for the Q&A. The GT is generated by GPT with human annotation. Reviewers would like to see more details on quality control, such as examples of bad cases, the preferences of 3 finance researchers during filtering, and the proportion of cases eliminated at each step. This would help assess the reliability of the GT.

**Questions:**

1. Why is it called MME-FINANCE? Is it meant to complement MME or does it have a different meaning for MME?
2. As a multimodal financial benchmark, do the model performance trends align with those from text-only benchmarks?
3. Apart from cognition tasks requiring financial knowledge, how different are the other tasks from mainstream benchmarks like MME? Do the evaluation results align with these mainstream benchmarks?
4. Could the authors provide more insights, such as how to improve the performance of such models after such a comprehensive MLLM review?

---

### Official Review · Reviewer_KL7R · 2024-11-04

**Soundness:** 3
**Presentation:** 3
**Contribution:** 3
**Rating:** 5
**Confidence:** 3

**Summary:**

Existing Multimodal Large Language Models (MLLMs) excel in understanding general natural images but face challenges in interpreting complex financial images, which require specialized knowledge and fine-grained reasoning. To address this gap, the authors propose the MME-FINANCE benchmark which is focusing on evaluating MLLMs' performance in open-ended financial Visual Question Answering (VQA). This benchmark includes over 1,000 VQA pairs across diverse financial scenarios, with tasks tailored to assess perception, reasoning, and cognitive abilities specific to finance. Using a multimodal evaluation approach aligned with human judgment, experiments on 18 mainstream MLLMs highlight their current limitations in finance, offering insights for further advancement.

**Strengths:**

The proposed benchmark is a new task for MLLM.

The efforts are awesome.

**Weaknesses:**

1. How did the expert revision conduct? More info should be enclosed. BTW, what is the "experts reversion" in Figure 2?
2. Using GPT4o to evaluation the performance of GPT4o is not fair. Cross-validation should be considered.
3. Lack of comparisons of Claude and Gemini, which is critical.

**Questions:**

See Weaknesses. More models should be considered for evaluation.

---

### Official Review · Reviewer_RUz1 · 2024-11-04

**Soundness:** 2
**Presentation:** 3
**Contribution:** 2
**Rating:** 5
**Confidence:** 3

**Summary:**

This paper proposes MME-FINANCE to evaluate the financial capability of MLLMs. MME-FINANCE consists of three levels, including Perception, Cognition and Reasoning. They collect data from financial images and manually check the automatically generated question and answer by GPT-4o. They conduct experiments to show the financial capability of MLLMs.

**Strengths:**

1. The category of this benchmark is novel. MME-Finance is the first benchmark to evaluate the financial capabilities of MLLMs.
2. The paper is well-written and easy to follow.

**Weaknesses:**

1. The evaluation process may not be reliable using GPT-4o as an evaluator shown in Fig. 4. The question and answer generated by GPT-4o still requires manual check. How to ensure the correctness using GPT-4o. Maybe the multiple-choice format is more reliable.
2. I think some classes of MME-FINANCE are not necessary, such as the capabilities of Image Caption, or OCR. There have been several benchmarks to evaluate these capabilities. Creating a new benchmark should focus on one specific capability. The hierarchical design of MME-Finance can be improved.
3. COT evaluation would enhance the comprehensiveness of experiments.

**Questions:**

No

---

### Official Review · Reviewer_Kib8 · 2024-11-05

**Soundness:** 2
**Presentation:** 3
**Contribution:** 2
**Rating:** 5
**Confidence:** 3

**Summary:**

- This work presents a multimodal understanding benchmark specifically for evaluating the capabilities of MLLMs in financial domains. With different image types and question focus, the benchmark could provide a comprehensive analysis of MLLM's capabilities in the financial domain.

  - Given the open-ended nature of some financial questions, the authors provide a novel evaluation strategy for better aligning with humans. By conducting an extensive evaluation of 18 MLLMs, insights about their ability in the financial domain are provided.

**Strengths:**

- This work investigates a rarely-explored setting for evaluating MLLM performance, particularly the financial domain. This focus on financial analysis is timely given the increasing complexity of financial data and the need for advancing analytical tools.

 - The proposed benchmark employs a rigorous multimodal evaluation method combined with image information, which enhances the alignment with human judgment.

 - The authors conducted thorough evaluations on 18 mainstream MLLMs, revealing critical insights into their limitations in processing financial tasks.

**Weaknesses:**

- While the benchmark covers various types of financial images (e.g., candlestick charts, statistical charts), it may not encompass all possible scenarios encountered in real-world finance. Given the relatively small number of image-question pairs (1171), the limitation could affect the generalizability of the findings to broader financial contexts.

 - The specialized context in the financial domain requires the reliance on expert annotators. Although the evaluation scores show a high relevance with human-annotated scores, this reliance may still introduce biases or inconsistencies based on individual interpretations of financial data.

**Questions:**

- What considerations lead to the selection of the six types of financial images included in MME-FINANCE, and are there plans to expand this scope to include additional types of financial data in future iterations?

 - Another concern is on the hallucination evaluations, as the number of evaluation samples is quite small. Considering that hallucinations could also happen in other capabilities and tasks evaluations, can you provide more details on how MME-FINANCE evaluates hallucinations in MLLMs?

---

### Note · Authors · 2024-11-15

I have read and agree with the venue's withdrawal policy on behalf of myself and my co-authors.